# Immunogenicity against the Omicron Variant after mRNA-Based COVID-19 Booster Vaccination in Medical Students Who Received Two Primary Doses of the mRNA-1273 Vaccine

**DOI:** 10.3390/vaccines10122102

**Published:** 2022-12-08

**Authors:** Hyemin Chung, Jongsung Lee, Kyungrok Minn, Jiyoung Lee, Soyoung Yun, Joung Ha Park, Min-Chul Kim, Seong-Ho Choi, Jin-Won Chung

**Affiliations:** 1Division of Infectious Diseases, Department of Internal Medicine, Chung-Ang University Hospital, Seoul 06973, Republic of Korea; 2College of Medicine, Chung-Ang University, Seoul 06974, Republic of Korea; 3Division of Infectious Diseases, Department of Internal Medicine, Chung-Ang University Gwangmyeong Hospital, Gwangmyeong 14353, Republic of Korea

**Keywords:** booster, neutralizing antibody response, Omicron, medical school students

## Abstract

We evaluated the immune response against the Omicron variant after mRNA-based COVID-19 booster vaccination in medical students. We prospectively enrolled medical students who received two primary doses of the mRNA-1273 vaccine. The neutralizing response and the SARS-CoV-2-specific T-cell response was evaluated. A total of 56 serum samples were obtained before booster vaccination. Nineteen students (33.9%) developed COVID-19 two months after booster vaccination. Of 56 students, 35 students (12 infected and 23 uninfected) were available for blood sampling four months after booster vaccination. In comparison with uninfected students, infected students showed a significantly higher level of SARS-CoV-2-specific IgG (5.23 AU/mL vs. 5.12 AU/mL, *p* < 0.001) and rate of neutralizing response (96.22% vs. 27.18%, *p* < 0.001) four months after booster vaccination. There was no significant difference in the SARS-CoV-2-specific T-cell response. Among 23 infection-naive students, the neutralizing response was significantly higher in those who received the mRNA-1273 booster than in those who received the BNT162b2 booster (69.07% vs. 26.43%, *p* = 0.02). In our study, booster vaccination with mRNA-1273 instead of BNT162b2 was significantly associated with a higher neutralizing response.

## 1. Introduction

The introduction of new vaccines against severe acute respiratory syndrome coronavirus 2 (SARS-CoV-2) offered hope that it would end the coronavirus disease 19 (COVID-19) pandemic. However, the emergence of the Omicron variant (B.1.1.529) of SARS-CoV-2, which possesses numerous mutations in the epitopes of neutralizing antibodies on the viral spike glycoprotein, has rapidly increased COVID-19 cases across the globe, escaping vaccine-induced immunity [1]. Although previous studies found that booster vaccines could elicit a neutralizing antibody response against the Omicron variant [2,3,4,5], there are limited real-world data on long-term humoral and cellular immunity against the Omicron variant after booster vaccination. 

Medical students are prehealthcare workers and likely to be at increased risk of exposure to SARS-CoV-2. It is vital to provide them with COVID-19 vaccines for protection and prevent COVID-19 outbreaks in hospitals. In South Korea, medical students had received two doses of the mRNA-1273 vaccine in June 2021 and an mRNA-based booster vaccine (mRNA-1273 or BNT162b2) in December 2021, according to the government’s approved schedule for vaccination. Some medical students in other countries have indicated that they were hesitant to receive COVID-19 vaccines because of concerns about vaccine side effects and distrust of public health experts [6,7]. Therefore, we aimed to evaluate the immune response against the Omicron variant four months after the third dose of a COVID-19 mRNA vaccine in medical students in this study. 

## 2. Methods

### 2.1. Study Design

This prospective study was conducted with medical students at Chung-Ang University in Seoul, South Korea. In December 2021, we recruited students who had received two primary doses of mRNA-1273 and were scheduled to be vaccinated with an mRNA-based booster vaccine. The interval between the two primary doses of the mRNA-1273 vaccine was 4 weeks. Blood samples were collected from the students before booster vaccination (five months after the second vaccination) and four months after booster vaccination. The students filled out questionnaires about sex, age, type of vaccines, date of vaccination, adverse reactions to vaccines, and their history of confirmed COVID-19. 

### 2.2. Measurement of SARS-CoV-2 Spike-Specific Immunoglobulin G

We used an anti-SARS-CoV-2 enzyme-linked immunosorbent assay (ELISA) kit (Euorimmun, Lubeck, Germany) to measure SARS-CoV-2 spike-specific immunoglobulin G (IgG), as previously described [8]. The microplate wells were coated with the recombinant structural protein of SARS-CoV-2. The results are shown as the measured optical density (OD) at 450 nm with responses expressed as arbitrary units per milliliter (AU/mL). Antibody titers greater than 1.1 AU/mL were interpreted as positive according to the manufacturer’s instructions.

### 2.3. Measurement of Neutralizing Antibody Response

Neutralizing antibodies were measured using the GenScript SARS-CoV-2 Surrogate Virus Neutralization Test (SVNT) kit (Genscript Biotech Corporation, Piscataway, NJ, USA), as previously described [8]. The kit indirectly detects potential SARS-CoV-2 neutralizing antibodies by determining the antibody blocking of the binding of the SARS-CoV-2 receptor-binding domain to the human host cellular receptor (angiotensin-converting enzyme 2). The percentage of neutralization was calculated as (1 − OD of sample/OD of negative control) × 100, and the recommended positive threshold was 30%. The test was modified to detect the neutralizing response against the receptor-binding domain of the Omicron variant (BA.1) by replacing the horseradish peroxidase conjugated recombinant receptor-binding domain fragment according to the manufacturer’s instructions. 

### 2.4. Measurement of SARS-CoV-2-Specific Cellular Response

After stimulation with the SARS-CoV-2 spike protein, IFN-γ production was measured to evaluate the SARS-CoV-2-specific T-cell response, using the SARS-CoV-2 Interferon Gamma Release Assay (IGRA) kit (Euroimmun, Lubeck, Germany), as previously described [8]. The cellular response was defined as the stimulated minus unstimulated IFN-γ peptide expressed as international units per milliliter (IU/mL). An IFN-γ response above 200 mIU/mL was defined as positive. The SARS-CoV-2-specific cellular response was evaluated using blood samples obtained from students four months after booster vaccination. 

### 2.5. Statistical Analysis

We used the chi-square test or Fisher’s exact test for categorical variables as appropriate, and Student’s *t*-test and the Mann–Whitney *U*-test for continuous variables as appropriate. The continuous variables are presented as median values with the interquartile range (IQR). The Spearman correlation test was used to analyze linear correlation. *p* values less than 0.05 were used to denote statistical significance. We used GraphPad Prism version 5.0 and Statistical Package for the Social Sciences (version 23.0; IBM Corp., Armonk, NY, USA) to perform all analyses.

## 3. Results

### 3.1. Study Population

Figure 1 shows the study flow chart. A total of 56 students who had received two primary doses of the mRNA-1273 vaccine were enrolled in December 2021. Of these students, 26 (46.4%) of them were male, and the median age was 24.0 years (IQR, 24.0–25.0). Before booster vaccination, 56 blood samples were obtained. The median interval between the date of the second dose of vaccination and the date of the first blood sampling was 148.0 days (IQR, 147.0–149.0). No one had a history of confirmed COVID-19 before booster vaccination. A total of 19 (33.9%) students developed COVID-19 after receiving the booster vaccine during the follow-up period (until April 2022). The median interval between the date of booster vaccination and the date of confirmed COVID-19 was 77.5 days (IQR, 73.3–79.5) Of the 56 students, 35 students (12 infected and 23 uninfected with COVID-19) were available for a second blood sampling four months after booster vaccination. The median interval between the date of booster vaccination and the date of second blood sampling was 119.0 days (IQR, 119.0–120.0).

### 3.2. SARS-CoV-2 Spike-Specific IgG

Overall, the median value of SARS-CoV-2 spike-specific IgG was significantly higher five months after the second dose of vaccination compared with that assessed four months after booster vaccination (6.39 AU/mL vs. 5.18 AU/mL, *p* < 0.001). Among COVID-19-infected students and uninfected students, the median value of SARS-CoV-2 spike-specific IgG (6.59 AU/mL vs. 6.22 AU/mL, *p* = 0.50) was comparable five months after the second dose of vaccination (two months before being confirmed with COVID-19). Infected students had a higher level of SARS-CoV-2 spike-specific IgG four months after booster vaccination (two months after being confirmed with COVID-19) compared with that of uninfected students (5.23 AU/mL vs. 5.12 AU/mL, *p* < 0.001, Figure 2).

### 3.3. Neutralizing Antibody Response against Wild-Type and OMICRON Variant

The neutralizing antibody response against the wild-type remained at a high level both before and four months after booster vaccination, and there was no significant difference in the neutralizing antibody response against the wild-type between infected and uninfected students (five months after the second dose of vaccination, 97.76% vs. 97.60%, *p* = 0.41; four months after booster vaccination, 98.31% vs. 98.27%, *p* = 0.38) (Figure 3). Among uninfected students, the median rates of the neutralizing antibody response against the Omicron variant were 31.45% before booster vaccination and 27.18% after booster vaccination (*p* = 0.71). Of 23 uninfected students available for a second blood sampling, 5 and 18 of them received mRNA-1273 and BNT162b2 as booster vaccines, respectively. Students who received the mRNA-1273 booster showed a significantly higher rate of neutralizing response compared with the rate of those who received the BNT162b2 booster (69.07% vs. 26.43%, *p* = 0.02, Figure 4); however, there was no difference in the incidence of breakthrough COVID-19 according to the booster type (mRNA-1273, 1/6 (16.7%) vs. BNT162b2, 11/29 (37.9%), *p* = 0.64). After receiving the booster vaccine and being confirmed with COVID-19, infected students showed a significantly higher neutralizing response against Omicron compared with the response of uninfected students (96.22% vs. 27.18%, *p* < 0.001, Figure 3).

### 3.4. SARS-CoV-2-Specific Cellular Response

The SARS-CoV-2-specific cellular response was evaluated using blood samples obtained four months after booster vaccination. Figure 5 shows the level of IFN-γ of infected and uninfected students four months after booster vaccination. The IFN-γ level was not significantly different between infected and uninfected students (563.69 mIU/mL vs. 344.26 mIU/mL, *p* = 0.21); similar results were obtained for the positive rate (91.7% vs. 69.6%, *p* = 0.22). There were no significant linear correlations between IFN-γ level and IgG level (*p* = 0.33), neutralizing antibody response against wild-type (*p* = 0.86), or neutralizing antibody response against the Omicron variant (*p* = 0.21) in uninfected students. Among uninfected students, the mRNA-1273 booster group and the BNT162b2 booster group showed similar results of median IFN-γ level (*p* = 0.45).

### 3.5. Vaccine Side Effects

The most common side effects associated with booster vaccination were myalgia and arthralgia (51.4%), injection site pain (51.4%), and general weakness (40.0%); on the other hand, injection site pain (76.8%), fever (73.2%), and myalgia and arthralgia (71.4%) were the most common side effects associated with the second dose of vaccination (Table 1). Fever, headache, chilling sense, general weakness, and injection site pain were significantly less common after booster vaccination compared with second-dose vaccination.

## 4. Discussion

In our study, the hybrid immunity induced by natural infection after third-dose vaccination was associated with a strong neutralizing response against the Omicron variant, which is consistent with previous findings [9,10]. Infection-naive students showed a low neutralizing response against the Omicron variant (27.2%) four months after booster vaccination, suggesting insufficient protection against the Omicron variant with the booster vaccine alone. Among infection-naive students, booster vaccination with mRNA-1273 instead of BNT162b2 was significantly associated with a higher neutralizing response against the Omicron variant four months after booster vaccination. 

Since the Omicron variant was first detected in December 2021 in South Korea [11], an Omicron-fueled surge started in January 2022, which peaked in March 2022 [12]. During this surge, around one-third of medical students experienced breakthrough COVID-19 despite booster vaccination in our study. The median rates of the neutralizing response against the Omicron variant and levels of SARS-CoV-2 spike-specific IgG prior to breakthrough infection did not differ significantly between infected and uninfected students. Social distancing rather than vaccine-induced immunity might be implicated in this breakthrough SARS-CoV-2 infection. In addition, among infection-naive students, the neutralizing response against the Omicron variant measured before booster vaccination was not significantly different from that measured four months after booster vaccination. Previous studies found that neutralization titers against the Omicron variant were reduced six months after receiving the booster, suggesting a waning effect [13]. Although we did not evaluate the neutralizing response of medical students immediately after receiving the booster vaccine in our study, our healthcare worker cohort in another study showed that the mean neutralizing response against the Omicron variant was 50.3% 8 weeks after booster vaccination [8]. The relatively low neutralizing response against the Omicron variant (27.18%) might be attributed to the vaccine’s waning effect. As the currently available booster vaccines offer limited immune protection against new mutant viral infections, novel vaccines might be needed. 

Neutralizing antibody responses are associated with blocking SARS-CoV-2 infection, while both humoral and cellular responses are associated with preventing progression to severe disease [14]. The booster vaccine showed a protective effect against hospitalization or death due to the Omicron variant infection [15]. Vaccine-induced cellular immunity is considered to stay largely intact [14]. In our study, there was no significant difference in T-cell response after booster vaccination between COVID-19-infected and uninfected students, and infected students all had mild symptoms. Vaccine-induced T-cell response might be maintained even in the absence of natural infection unlike neutralizing response against Omicron variants. However, we did not measure IFN-γ levels before booster vaccination and other cytokines’ levels, including interleukin-2 in our study. Therefore, to understand T cellular response properly, further studies are needed. Moreover, we could not find any significant correlation between the IFN-γ level and SARS-CoV-2 spike-specific IgG level, neutralizing activities against wild-type, or neutralizing activities against the Omicron variant in uninfected students. There was no significant difference in the median IFN-γ level according to the booster type. Because we did not perform anti-Np test to confirm a history of COVID-19 infection, asymptomatic COVID-19 infection could be one of the possible explanations for a high level of IFN-γ in uninfected students.

The mRNA-1273 vaccine has been reported to be associated with myocarditis or pericarditis, especially in young adults [16]. Therefore, the South Korean government recommended the mRNA-1273 vaccine for adults over the age of 30 and as a booster vaccine for adults over the age of 18 since November 2021. However, medical students had received two primary doses of mRNA-1273 before the government’s proposed changes. They could choose the type of booster vaccine (mRNA-1273 or BNT162b2). There were no serious vaccine-associated side effects including myositis and pericarditis in our study, and side effects associated with booster vaccination appeared to be less common than those associated with second-dose vaccination. Among infection-naive students, booster vaccination with mRNA-1273 instead of BNT172b2 was significantly associated with a higher neutralizing response against the Omicron variant. A British team reported that the rates of vaccine effectiveness against the Omicron variant were 64.9% and 66.3% at 2–4 weeks after booster vaccination among patients who received mRNA-1273 (primary) with mRNA-1273 (booster) and those who received mRNA-1273 (primary) with BNT162b2 (booster), respectively [17]. There is a lack of data on the long-term neutralizing antibody response according to mRNA-1273 or BNT162b2 booster vaccination in young persons who received two primary doses of mRNA-1273. Nevertheless, mRNA-1273 as a booster vaccine may be safe and more effective than BNT162b2 for young persons who were administered a primary series of mRNA-1273.

There were several limitations in our study. First, this study was limited by the small sample size in a single center. In particular, only five uninfected students who received mRNA-1273 booster vaccines were included in the analysis to compare neutralizing response according to the booster type. Second, some students were lost to follow-up for blood sampling four months after booster vaccination. Third, we confirmed the history of COVID-19 through an in-depth interview but not an S1-specific IgG test. Asymptomatic or paucisymptomatic COVID-19 students could be misclassified as uninfected. Fourth, the SVNT kit we used to measure neutralizing antibody responses could not detect the neutralizing antibodies that targeted other regions by different neutralizing mechanisms. A further study by a cell-based neutralization assay is needed.

## 5. Conclusions

The currently available booster vaccines may not be sufficient to protect against infection caused by the Omicron variant. Novel vaccines with a lower waning effect, which can confer immunity against new variants, are needed. 

## Figures and Tables

**Figure 1 vaccines-10-02102-f001:**
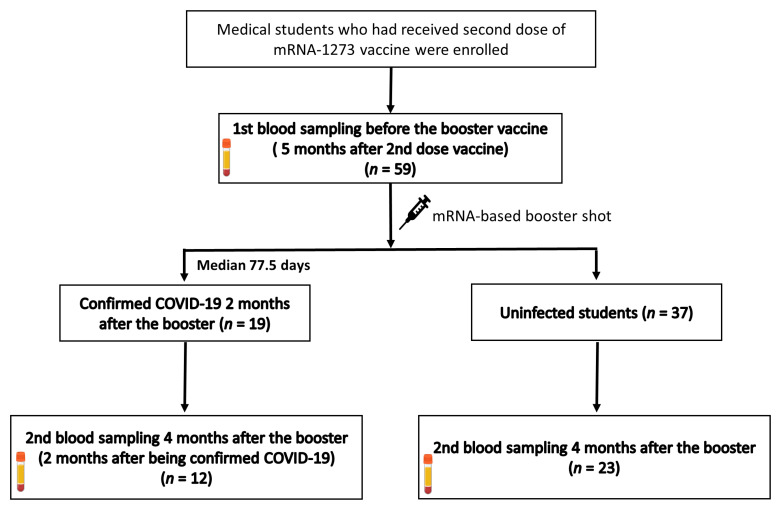
Study flowchart.

**Figure 2 vaccines-10-02102-f002:**
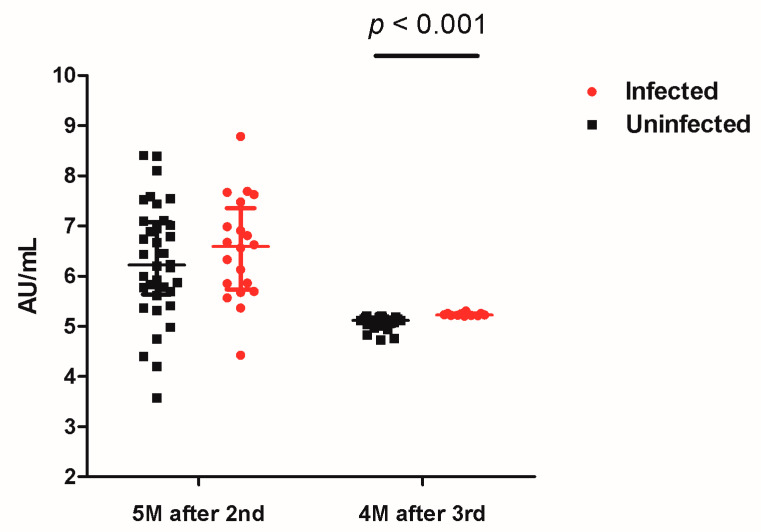
SARS-CoV-2 spike-specific immunoglobulin G (IgG) five months after second-dose vaccination and four months after booster vaccination in COVID-19-infected and uninfected students.

**Figure 3 vaccines-10-02102-f003:**
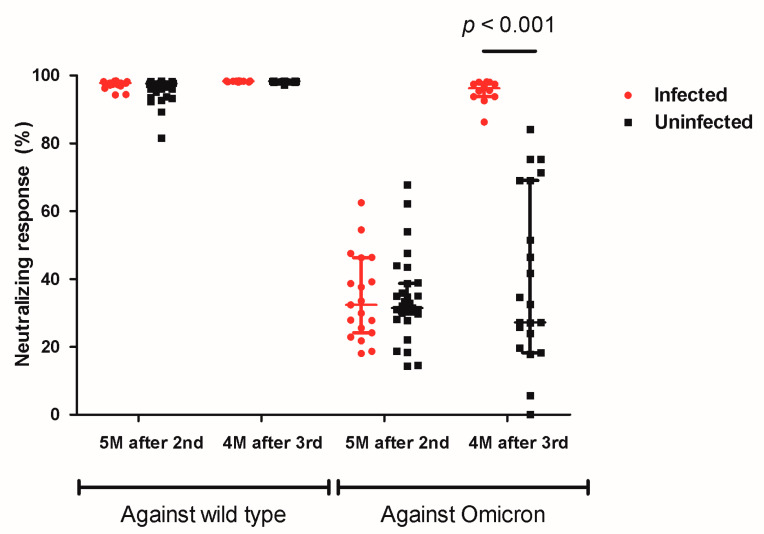
Neutralizing response against the wild-type and Omicron variant in COVID-19-infected and uninfected students.

**Figure 4 vaccines-10-02102-f004:**
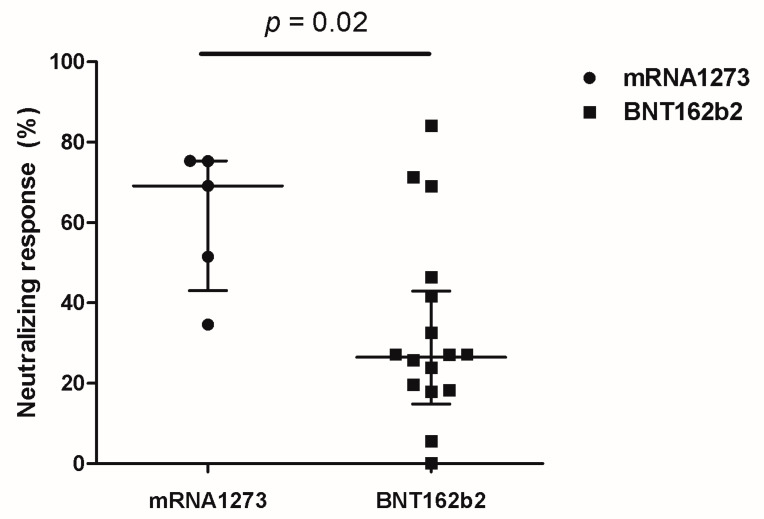
Neutralizing response against the Omicron variant in COVID-19-infection-naive students according to the type of booster vaccine.

**Figure 5 vaccines-10-02102-f005:**
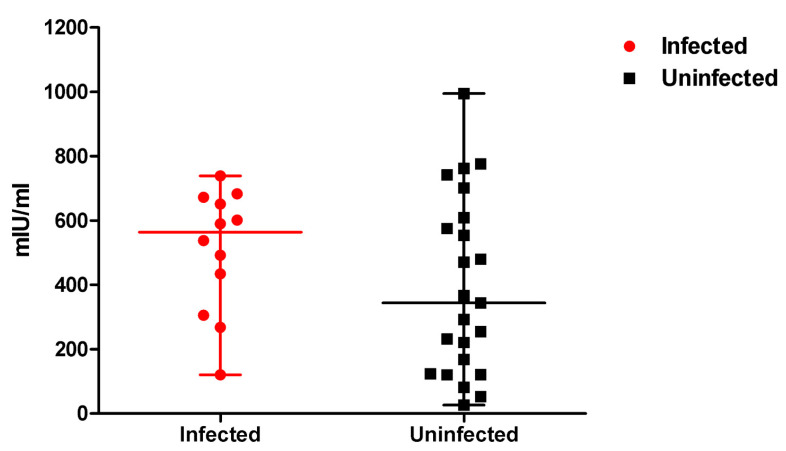
SARS-CoV-2-specific T-cell response after booster vaccination in COVID-19-infected and uninfected students.

**Table 1 vaccines-10-02102-t001:** Side effects after the second dose of vaccination and booster vaccination in medical students.

Side Effects	Second Dose (*n* = 56)	Booster (*n* = 35)	*p* Value
Fever	41 (73.2)	8 (22.9)	<0.001
Headache	26 (46.4)	8 (22.9)	0.02
Chilling sense	36 (64.3)	9 (25.7)	<0.001
Nausea	6 (10.7)	2 (5.7)	0.71
Vomiting	1 (1.8)	0	>0.99
Diarrhea	1 (1.8)	2 (5.7)	0.56
Myalgia arthralgia	40 (71.4)	18 (51.4)	0.054
General weakness	36 (64.3)	14 (40.0)	0.02
Injection site pain	43 (76.8)	18 (51.4)	0.01
Injection site tenderness	12 (21.4)	4 (11.4)	0.22
Injection site swelling	11 (19.6)	5 (14.3)	0.51
Needs nonsteroidal anti-inflammatory drugs or acetaminophens	44 (78.6)	21 (60.0)	0.056

## Data Availability

Not applicable.

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
