# Peer review of "Immunogenicity against the Omicron Variant after mRNA-Based COVID-19 Booster Vaccination in Medical Students Who Received Two Primary Doses of the mRNA-1273 Vaccine"

_vaccines, 2022, doi:10.3390/vaccines10122102_

Round 1
Reviewer 1 Report
The authors confirm the stronger immunity in hybrid immunity than vaccination alone in both cellular and humoral immune response. The author also demonstrated the higher neutralisation activity in mRNA recipients than Bnt162b2. I have only few comments as following:
1. Please add the interpretation of positivity of IgG measurements I understand that titres greater than 1.1 AU/mL were considered to be seropositive.
2. Any anti-Np had been done or could be performed? Since the infection was solely based on medical history, not routine surveillance. Vaccinated individuals whom infected with omicron, somehow, may be asymptomatic. (Vaccines (Basel). 2022 Jul; 10(7): 1049.) Since the asymptomatic rate could be as high as 30% according to the referred article.
3. Uninfected subjects who had very high IGRA testing is worth to a little bit more be discussed or explored. Any correlation between IGRA test and IgG level or neutralizing activities in these patients?
Reviewer 2 Report
In this manuscript, Chung et al. evaluated the immunogenicity induced by the COVID-19 booster of two different vaccines (mRNA-1273 or BNT162b2) after two primary doses in medical students in South Korea. The authors assessed antigen-binding IgG antibodies, neutralizing antibodies, and T cell responses against SARS-CoV-2 spike protein at different timepoints before and after the boosting immunization and revealed that infection induced strong IgG responses, especially against Omicron variant, while mRNA1273 as the booster may induce higher neutralizing response compared with BNT162b2. The manuscript is well-written, however, the novelty is relatively low, so it may be better if the authors can emphasize the novelty and the significance of the study, as well as addressing the following comments.
The authors only tested the antibody responses against wild-type and Omicron variants in this study, what about other variants? And is the Omicron variant indicated here referring to BA.1? Can the authors also include the analysis against BA.2 and other Omicron variants since now the dominant VOC is not BA.1 anymore and other Omicron variants showed more extreme escape mutations.
For the measurement of neutralizing antibody responses, the authors utilized the SVNT kit by determining antibody blocking RBD binding to ACE2, however, this kit cannot detect the neutralizing antibodies that target other regions by different neutralizing mechanisms such as conformational trapping etc. so it may be better if the authors can measure the neutralizing antibody responses by a cell-based neutralization assay or other approaches.
For the cellular response analysis, the authors measured the IFN-γ production stimulated by spike protein, however, other cytokines may also need to be analyzed such as IL-2, and there are commercially available kits for this. It should be necessary to investigate different aspects for the T cell responses. In addition, it may be necessary to include T cell response analysis before booster as a control for the comparison.
In figure 1, in the left branch of the flow chart the authors wrote “confirmed COVID-19 2 months after the booster (n=19)”, did all the 19 students get infected at exactly 2 months after boost?
When comparing between booster of mRNA1273 and BNT162b2 in non-infected donors, there were only 5 samples for the mRNA1273 group, in contrast to 18 samples for BNT162b2 group. This sample size difference especially the small sample size for the mRNA1273 group may possibly affect the statistical analysis for the comparison.
Some sentences need refinement, for example, in line 122, “both before vaccination and four months after booster vaccination” should be “both before and four months after booster vaccination”; line 35-136, “higher neutralizing response” should be “higher neutralizing response against Omicron”.
Round 2
Reviewer 2 Report
The authors addressed the points one by one in the revision. Several assays (including assays against other SARS2 variants, IL-2 cytokine analysis, and cell-based neutralization) were not available in the lab to make the conclusions more solid, but the authors did include those in the discussion of limitations. Therefore, the manuscript can be accepted.